# AR in the Architecture Domain: State of the Art

**Michele Russo** 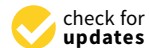

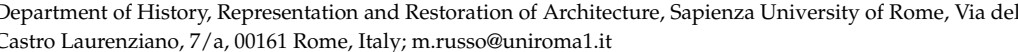

Department of History, Representation and Restoration of Architecture, Sapienza University of Rome, Via del Castro Laurenziano, 7/a, 00161 Rome, Italy; m.russo@uniroma1.it

**Featured Application: This article proposes an AR overview in Architecture, highlighting the main methodological steps in the AEC and educational fields, encouraging a unified view of a fragmented subject towards a democratization of AR in the domain.**

**Abstract:** Augmented reality (AR) allows the real and digital worlds to converge and overlap in a new way of observation and understanding. The architectural field can significantly benefit from AR applications, due to their systemic complexity in terms of knowledge and process management. Global interest and many research challenges are focused on this field, thanks to the conjunction of technological and algorithmic developments from one side, and the massive digitization of built data. A significant quantity of research in the AEC and educational fields describes this state of the art. Moreover, it is a very fragmented domain, in which specific advances or case studies are often described without considering the complexity of the whole development process. The article illustrates the entire AR pipeline development in architecture, from the conceptual phase to its application, highlighting each step's specific aspects. This storytelling aims to provide a general overview to a non-expert, deepening the topic and stimulating a democratization process. The aware and extended use of AR in multiple areas of application can lead a new way forward for environmental understanding, bridging the gap between real and virtual space in an innovative perception of architecture.

**Keywords:** augmented reality; architecture; knowledge workflow; digital content; MAR; AEC; modeling process; virtual representation; education

## 1. Introduction

Augmented reality (AR) is a research domain in ongoing transformation due to vast demand and developments in ICT. In recent years, the diffusion of AR has been mainly due to technological and social reasons. As far as the former is concerned, the extensive massive digitization of 2D and 3D data contributes to the definition of increasingly large repositories of digital information. Moreover, the leap forward in the hardware and software capacities devoted to this domain is crucial [1]. Both conditions foster the application of AR, providing billions of users in several fields. In addition, this tool supports the growing desire to access and interact with digital information everywhere. The actual society is constantly engaged in multiple communication data flows that intersect and overlap. If managed properly, this information can significantly support and stimulate the understanding and interpretation of a complex reality. In this passage, the interaction between users, real data, and digital data becomes crucial, and AR has a central role to play.

In the past 20 years, AR research and training experiences have been carried out in several fields, including medical and biomedical, robotic, industrial, educational, and entertainment, to name but a few [2–4]. This article explores the architecture domain, because it represents an experimental area with wide margins for development. Such potential is due to architecture's pervasive diffusion, cultural content, social value, and the massive number of actors working in this realm. The European Union (EU) strongly supports these purposes. Starting from the Digital Agenda (2010), many efforts have been carried

out to digitize cultural heritage (DCH), redefining new knowledge and communication paths about architectural content. Almost all the most recently funded projects on digital cultural heritage (INCEPTION; i-MareCulture; Time Machine; ViMM) concerned AR use. In the 2021–2024 Agenda, one of the pillars is devoted to empowering people with a new generation of technologies, contributing to the realization of the European "Digital Decade". This will promote the built domain's accessibility, interoperability between institutions and users, preservation of national identities, territorial belonging, and cultural awareness. The diffusion and multi-generational communicative capacity of AR will facilitate achieving these goals, suggesting an innovative architectural experience [5] supporting design and space communication [6].

The application of ICT for cultural content reading, understanding, and communication is already a consolidated practice in the cultural computing domain [7]; it deepens digital applications in the arts and social sciences to represent, enhance, and extend knowledge, introducing new creative processes [8,9]. The availability of digital data, the requirements of access to knowledge, and the process optimization purposes foster the built domain as an ideal field for AR experimentation. Architecture is defined by several passages, from design to building best practices, analysis, monitoring and interpretation activities, open access to knowledge, and promoting heritage. Moreover, architectural education has a central role in strengthening personal and collective culture, and the correct use of the built heritage. The observation of architectonic space introduces both the importance of the observer's position and the spatial relationship between objects and the environment. Thus, AR can have a central role in cognitive terms, preparing the observer for a more conscious use, management, and design of built space.

Several aspects converge to form the AR built domain (Figure 1), from workflow design (AR Project) to digital content creation (virtual content), from system creation (AR System) to platform definition (AR SDK) and the evaluation of the cultural/education experience (experience feedback). The transdisciplinary meaning is highlighted by the preferential relationship with some ERC fields according to the 2021/22 panel. The selection proposed in Figure 1 is a qualitative analysis based both on the contents and descriptions of the different ERC panels, and on the author's experience. There is, therefore, the awareness that many of the fields not mentioned may play a role in this tree structure. However, the scheme intends to show the fragmentation of the disciplinary fields involved in AR for built architecture for skills and research content. The cultural, social, and content aspects are mainly supported by the social sciences and humanities (SSH). In contrast, hardware, software, and algorithmic aspects are analyzed more in the physical sciences and engineering (PE). In the SSH domain, there is a more significant sector fragmentation related to the topic. SH2 explores the issues of democratization of communication channels, and their impact on society. Studies devoted to the real environment, educational models, and transfer of skills are referred to as SH3. The content activities and construction related to the cultural studies and dissemination of past and existing tangible and intangible artifacts are referred to as SH5/SH6, up to local and geographical data management (SH7). Moreover, the PE6 sector is prevalent in the macro-sector PE, devoted to computational systems, guided and autonomous algorithmic methods, and user interface (HCI). It is emphasized that these connections and their different roles and weights within the methodology are strictly outlined in AR for architecture. This fragmentation (Figure 1) involves numerous studies, which mainly deepen acquisition, tracking, visualization, and interaction aspects, suggesting a technological and algorithmic solution [10–13]. Only a few are devoted to explaining the whole workflow in the built realm [7,14], or focused on transferring knowledge [15]. This lack may depend on the peculiarities and problem-solving approaches of the research in question, which can develop specific aspects without managing the subject as a whole. This specificity promotes the interest of experts in the domain, adopting a language and terminology consistent with the field, limiting the transdisciplinary dissemination of content, and promoting a watertight circuit that feeds itself.

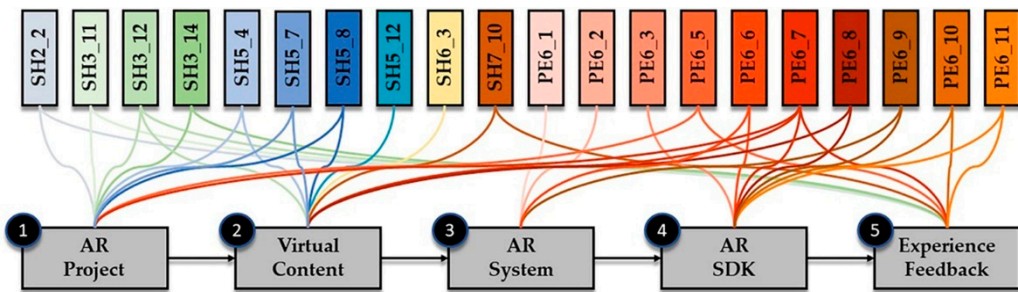

**Figure 1.** Diagram of the relationships between ERC sectors and the AR chain in the architecture domain.

This review aims to address an overview of AR in the architecture domain, encouraging a transverse and transdisciplinary reading of the entire pipeline and its specificity. The article follows a holistic approach, collecting the different experiences conducted in the field with the purpose of:

- Defining AR framed in the path of architectural knowledge;
- Organizing the AR pipeline, highlighting the main aspects of the chain;
- Outlining the state of the art of AR in the architecture domain;

The in-depth study of these topics will be developed according to the following sections: Section 2 is dedicated to the definition of AR and its location within the reality–virtuality continuum, making some initial critical considerations. From Section 3 to Section 7 are discussed the main methodological steps. Specifically, in Section 3 are presented the main criteria to build an AR project, analyzing each inference with the architecture domain. Section 4 is devoted to digital content creation, explaining the levels of iconicity and data typology. Section 5 traces the primary HW and SW elements in an AR system, from the different devices to the tracking and registration systems, up to the HCI. Section 6 is dedicated to an in-depth analysis of the currently existing AR platforms related to the world of architecture, deepening the different types, purposes, and potentialities. Section 7 addresses the issue of feedback on the AR experience—a critical matter, especially in education, but necessary to improve the experience offered by AR. Finally, Section 8 is dedicated to the state of the art of AR research in Architecture, verifying the different topics related to the built domain. Some brief critical conclusions on the in-depth study close the review.

## 2. AR in the Real-Virtual Continuum

The augmented reality sphere relates to both the real environment and the virtual world. AR is placed within a well-known spatial interval (continuum) between real and virtual (Figure 2), named X-reality (XR), which contains real environment (RE), augmented reality (AR), augmented virtuality (AV), and virtual reality (VR) [16].

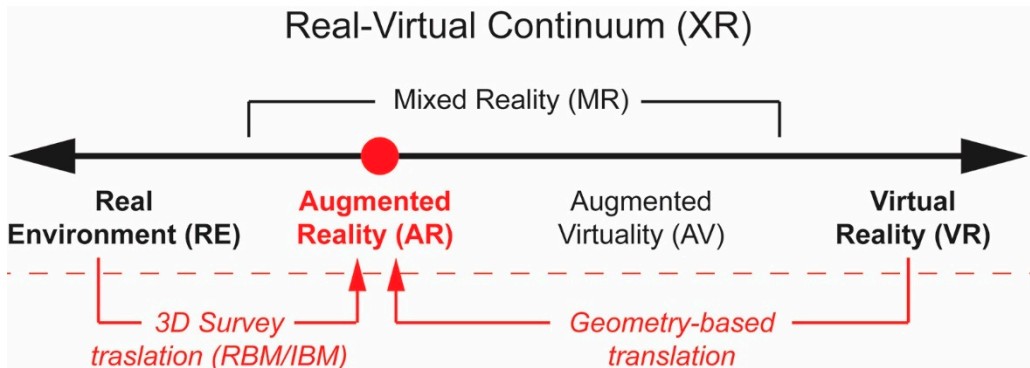

**Figure 2.** Milgram and Kishino's scheme, reinterpreted and adapted for the architecture domain.

Mixed reality (MR) describes the domain between AR and VR. AR's position in the continuum, close to physical reality [17], introduces virtual information into an existing reality via devices. This integration can be additive, layering information that does not exist, or subtractive, covering or deleting parts of the real world. The first case is referred to as AR, while the second as mediated reality. The latter, like AR, can be placed close to the real environment in the continuum towards AV. Within architecture, the transformation of real or virtual data towards the AR environment is supported by well-established methodologies discussed in Section 4. The connection between RE, AR, and VR (Figure 2) is made through 3D surveying and modeling methodologies that allow for the digitization of reality, such as reality-based modeling (RBM) or image-based modeling (IBM). At the same time, parametric models are artificially built through a geometry-based approach [18].

In the past decade, the definitions of AR highlighted and changed some specific elements or relations involved in the process. First, Milgram and Kishino in 1994 defined AR as: "*any case in which an otherwise real environment is "augmented" by means of virtual (computer graphic) objects.*" This definition pointed out the secondary role of AR, as an integrative medium to better understand the real environment, or a virtual reality connected to it [17]. This meaning was modified three years later in Azuma's definition: "*A system that combines real and virtual content, provides a real-time interactive environment, and registers in 3D*". [19]. The new definition was focused on the real-time representation, interaction, and its location in three-dimensional space. The "ethical" aspect of reality integration and multisensory approaches disappeared, stating the priority of the visual system over the other senses. In the following decades, this definition and the consequent taxonomy were expanded and integrated several times. AR is a system capable of improving the understanding of the world through the integration of virtual information, enlarging immersion and environmental knowledge [9,20,21]. The main AR characteristics (Figure 3), according to Azuma's definition and subsequent studies [22]. Outlined in the architecture domain, must present the following aspects:

- It must merge the real architectural environment, or a portion of it, with virtual data in a common visualization platform on the device;
- It must work in real time, reaching a synchronous update between the movements in the real space and the simulated virtual one;
- It must allow a high level of engagement, not constrained to a single point of view, moving freely in the architectonic space or around it;
- It must allow a variable level of interaction between the users, the real world, and the virtual contents.

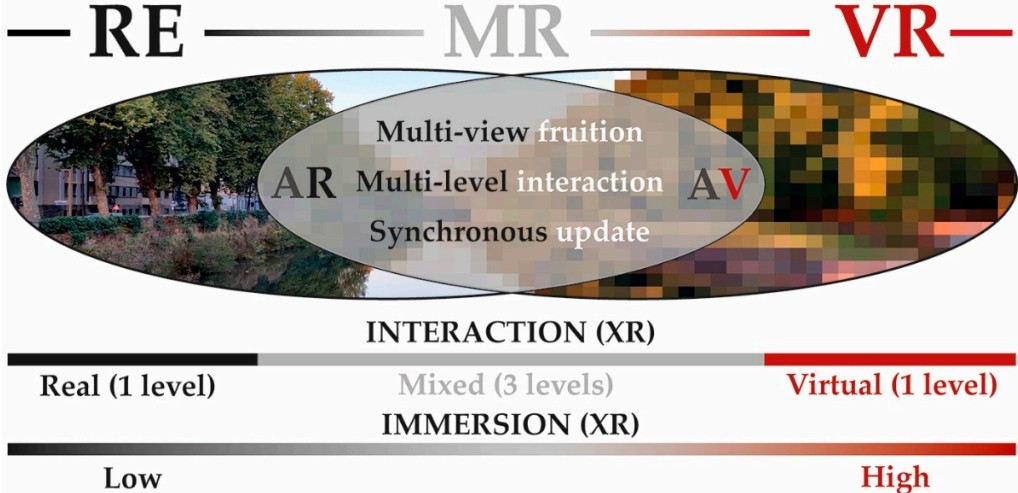

**Figure 3.** Diagram with the main features of AR, and its relationship to the real and virtual worlds.

The relationship between AR and the wider XR allows for the understanding of its interaction and immersion value [7]. Virtual reality (VR) permits a complete immersion in a digital world, presenting an environment disconnected from the real one. However, it can represent an interpretive reproduction [3], integrated with perceptual stimuli such as visual, auditory, and tactile stimuli to simulate the experience of a real environment [23]. Interaction occurs only with virtual information through different devices. In a symmetrical position to AR, augmented virtuality (AV) enriches the virtual world with scenes from the real one, enhancing the sense of presence and interaction with an artificial environment [24]. The domain between AR and VR is occupied by mixed reality (MR), in which real and virtual contents coexist, proposing an integrated, flexible, immersive, interactive platform for experience enhancement [25,26]. Interaction, in this case, is mixed, as it works on three different levels: At the real level, it occurs with body movement in the real space. At the virtual level, it happens only with virtual elements visualized in devices. The third level connects the real and the virtual levels by manipulating elements such as markers. [27]. All these environments (Figure 3) are characterized by different levels of immersion (i.e., immersive, semi-immersive, or non-immersive) depending on the type of device and visualization. The reduced field of view and precise tracking requirements of devices for AR visualization lead to the consideration of AR as a low-immersive representation technique. In general, the integration between personal devices, sensors, and large projection screens justifies increasing the level of immersion towards VR, but losing the connection with reality, which is very useful in the architecture domain.

### 3. AR Project

The AR design defines the starting point of the whole chain, decisive for the identification of both the main AR characteristics and its development. It is relevant to evaluate some aspects to prepare a suitable design path (Figure 4). Sidani [28] presents a helpful example, although focused only on the AECO world. First, the specific domain must evaluate either the content conveyed or the boundary conditions. Moreover, the primary purpose of AR must be clarified, as well as the final users. Based on these aspects, the outdoor or indoor AR conditions must be evaluated. Finally, it is crucial to assess the characteristics of the digital data merged with reality, the connection, and the capacity to increase contextual reading and understanding.

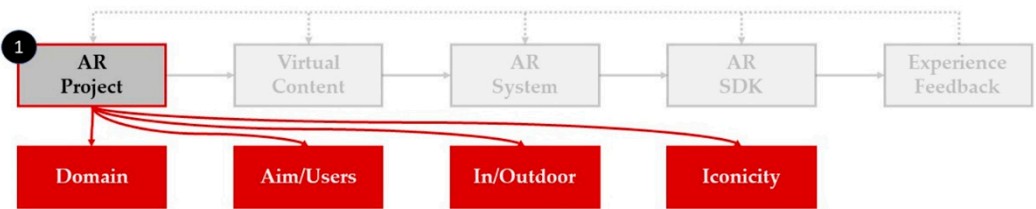

**Figure 4.** Block diagram of the main features of the AR system design.

### 3.1. AR Domain

A precise classification of this domain is complex since there are infinite hybridizations. Therefore, many variables characterize the AR system. The architectural complexity of the analyzed case and the boundary conditions of AR application [29] are meaningful. For the former, the virtuality–reality overlap is connected to the scale of visualization and the variable level of detail. These aspects intervene slightly on 2D data content. Instead, 3D information usually claims coherence with the visualized real scene, unless using out-of-scale architecture close-ups of specific details. The multiscale geometry is one of the main factors contributing to defining the architecture's complexity [30]. Since AR with 3D models works to visualize geometry in space, this characteristic must be carefully evaluated in an AR project, planning actions to cope with higher realization complexity. A complex subject allows the extension of AR data access in a contextual or unconstrained

way. Simultaneously, simple architecture with low-scale variation presents a reduced potential in terms of visualization, but a simpler realization (Figure 5).

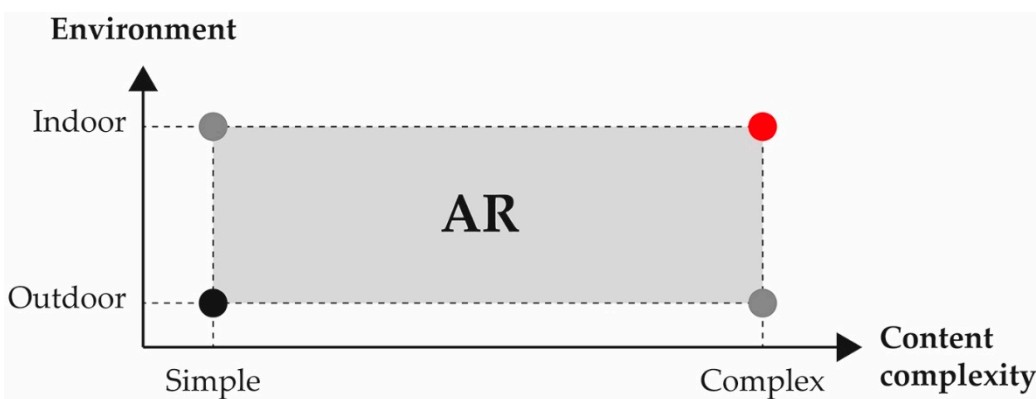

**Figure 5.** Graphical diagram with some domain variables.

The AR application context refers to the registration conditions of the real scene and the definition of the point of interest (POI), both connected with the subject peculiarities [31]. The two main conditions of AR application—outdoor or indoor—involve different content definitions and image/video acquisition choices. An indoor AR generally presents a reduced ambient light, making it much more challenging to identify the scene and track the system. Moreover, indoor POIs are often limited by the reduced area for data recognition in the architectonic volume, raising the AR application complexities. On the other hand, outdoor AR can easily foresee multiple POIs from the ground or from elevated positions. There is no correlation between architecture size and POIs, which instead depend on the storytelling designed for AR and the amount of digital data available to superimpose on the real environment.

### 3.2. AR Aim and Users

A careful evaluation of the AR system's communication aims and the users' target is fundamental in AR projects. The former identifies different levels of application, which can be expanded in several architectonic areas. It is interesting to highlight that the topic of architectural communication through AR is often deepened in the starting aim or the result of the research, but it is not self-consistent. Moreover, the word "communication" is frequently replaced by valorization and promotion terms. In general, the attention is focused on storytelling, strongly linked to the objectives of the project and the technology used [32]. A second aspect is related explicitly to AR's communicative and collaborative capacities [33], outlined in the architecture domain [20].

Suppose that the communication aims are set concerning the final users; in that case, it is appropriate to speak of levels of knowledge transfer, understood as clustering between groups that present homogeneity in terms of relationships and content, according to a qualitative analysis. The topic of knowledge transfer is extensive and related to multiple variables, from content creators to users, passing through the medium of communication and the environment in which it takes place. An interesting state of the art in this sense is offered by Becheikh et al. [34]. This division highlights the three levels of knowledge transfer: reception, adoption, and utilization of knowledge. These three areas can be expanded and outlined in architecture, suggesting a simplified version based on the author's experience in the domain. The main aims are clustered by the level of depth and the content complexity, as follows:

- Inform, promote, guide (Level I of knowledge transfer);
- Teach, learn, know (Level II of knowledge transfer);
- Design, interpret, understand (Level III of knowledge transfer).

In this suggested hierarchical structure (Figure 6), the first level represents the synthetic and simplified information, reaching the widest audience of users for a purely informative purpose without feedback. This level can be applied to most of the activities and application domains. An example is the cultural learning domain [35], which combines cultural promotion, engagement, and all those applications related to architecture, engineering, and construction (AEC) to support use, assembly, and preliminary analysis. In the second level, the virtual data must be designed to show concepts and calibrated for a more limited user, providing a feedback system on the AR used and the knowledge acquired. Here, AR communication is not a one-way information process, but aims to take root in the user, integrating with previous knowledge. To this level, all the educational paths related to architecture are mainly outlined in the teaching domain [36], construction training for public and private users, and tourist routes that provide specific learning tours. The last level (Figure 6) is devoted to exploring content that is otherwise challenging to understand with traditional communication channels. It extends already established knowledge and increases awareness in the architecture domain. The AR audience is very restricted by the cultural background suitable for understanding and using the complex content transmitted. The communication includes a first learning step enriched by a system of interaction that assesses the level of knowledge and verifies the concepts transferred. To this level, all users carry out an activity that requires a preliminary understanding and interpretation of the artifact before starting paths of built management and transformation. In general, target refers to users employed in the design sector, from building redevelopment to restoration, conservation, and research paths.

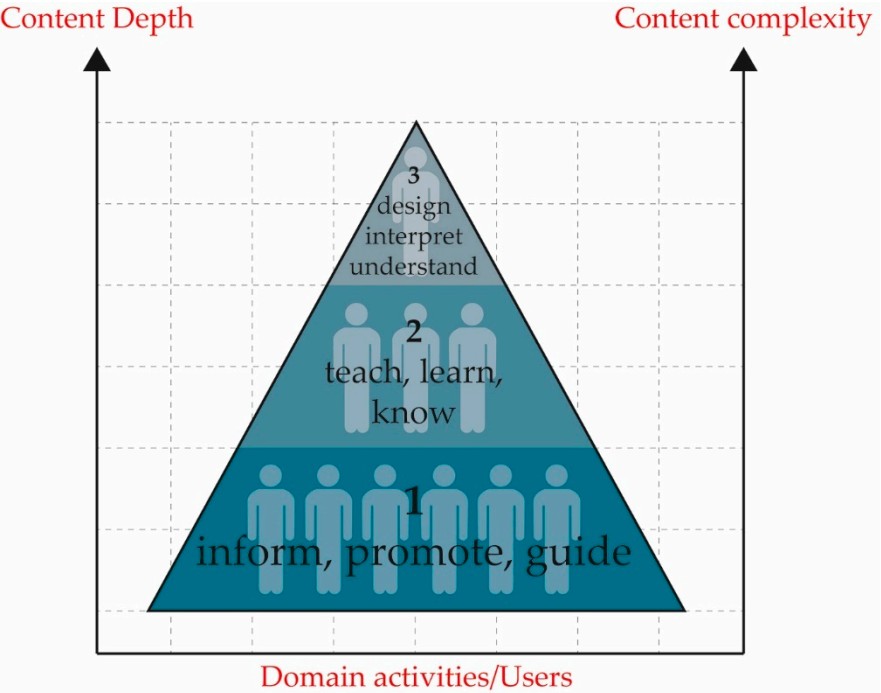

**Figure 6.** Graphical diagram with the AR aims related to content depth/complexity, and referring to the domain activities/users of the AR application.

### 3.3. Outdoor/Indoor Applications

The third aspect in the AR design chain is the boundary condition of application, which defines the suitable AR application. Starting from the subdivision given by [3] into five categories according to the variables indoor/outdoor and fixed/mobile devices, the division may be simplified into two macro-categories: outdoor and indoor AR. The outdoor AR system is the most frequently used because it can exploit the technological components of smartphones—the most popular AR device in the world. It is mainly

based on a markerless or mixed tracking system, portable displays, and tangible interfaces [37]. In some cases, it is possible to see the use of optical HMDs and collaborative interfaces [38,39]. Some research has explored and tested the use of these outdoor systems in the architecture/archaeology domains, such as [40–49].

Moreover, indoor AR applications with marker-based or markerless tracking and transparent HMDs, spatial or handheld displays, and tangible, collaborative, hybrid interfaces are possible. Internal systems do not need a GPS signal, but if the display is an HMD, the system could use inertial sensors to track the user's viewpoint. There are examples of markerless tracking for positioning along indoor pathways [50,51] or visual tracking to improve knowledge of cultural content [52]. Other research on the use of AR for indoor cultural content delivery can be found in [51,53–56].

### 3.4. AR Iconocity

The correct definition of AR virtual content assumes a decisive role in the suitable application of the vision system and the achievement of its purpose. In particular, it is crucial to define its simplification and integration levels compared to real examples, according to the subject's characteristics and the AR's aim (Figure 6). In such a sense, it is helpful to deepen the meaning of the virtual model related to AR. Recalling the theory of Abraham Moles [57], the models can present different levels of reality simplification, known as the "level of iconicity". A textual content, for example, presents a high level of simplification and iconicity in the scene description; it hints at human vision with integrative descriptions. The POI hotspot introduction is prevalent both for the simplicity of execution and for the independence of the visualization scale.

Moreover, all 2D visual multimedia contents are considered in AR to be "visual-based frames", regardless of the content—even if the level of iconicity is very variable within them. An architectural drawing, for example, is the result of an interpretative process, moving from the abstract (sketch) to accurate or executive drawings in the process of progressive refinement. Similarly, an image can be defined by a schematic representation of reality or a photograph, the closest representation of reality. Even 2D frames usually maintain this peculiar independence from the scale of representation, as well as the textual contents. The only exception is the superimposition between photographs or drawings and portions of the real architecture [58,59].

Finally, the 3D virtual models are the contents with the higher layered information, required to reach a dimensional and positional coherence with the real scene. In addition, there is a considerable difference in iconicity between the interpretive models and the reality-based ones (Figure 7). Images and 3D reality-based models define the lowest level of iconicity and the highest adherence with reality, depending on the instrumental characteristics and the survey methodology applied to acquire 2D/3D data.

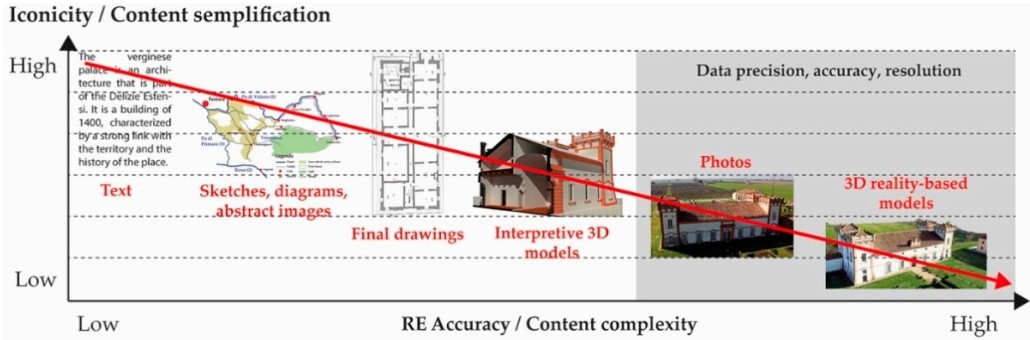

**Figure 7.** Graphic diagram showing the relationship between the iconicity of digital content and its descriptive simplification/complexity.

## 4. AR Virtual Content

Content creation (authoring) is central to the AR process, even if many articles refer to technical problems and development in hardware or software issues. The content must be coherent with the context (contextual awareness), following the three different AR aims (Figure 6), and meeting the communicative/educational goal while respecting a specific type of user. Moreover, different levels of virtual content can be associated and defined by different construction methodologies and relationships with the real context. The text strings present a low level of complexity that requires content validation and placement in the real context with POIs [31]. Even digital images, acquired by a camera or post-produced, show a spatial discrepancy between the 2D content and the 3D real space, using POIs unless the image is superimposed on reality according to a specific perspective. AR visualization projected on a plane display is suitable and consistent with 2D digital data representation.

Conversely, 3D models are different because they must define their state in the digital and real space, behavioral rules, relationships, and interactions [23]. For this purpose, there are different types of models and modeling methods. The primary division consists of reality-based models (RB data) and virtual-based ones (VB data) [60]. In 3D architectural modeling, the distinction between those models plays a key role, along with the model generation methodologies and issues addressed (Figure 8).

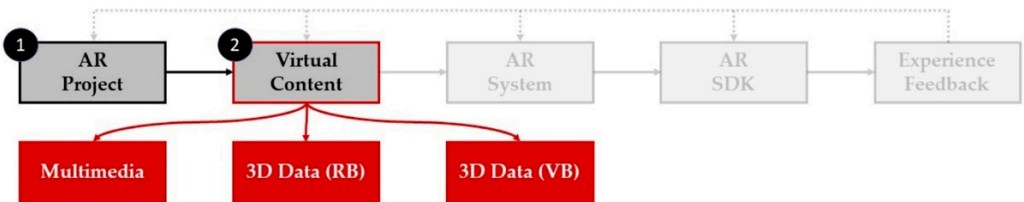

**Figure 8.** Block diagram of the main features of the virtual content creation.

### 4.1. Reality-Based 3D Models

AR in the architecture domain is strongly connected to the digitization process (Figure 9). In the past 20 years, the development of 3D acquisition methodologies has allowed for an enormous increase in heritage digitization. The digitization activity is central in supporting the protection, maintenance, conservation, and promotion activities; It allows for the democratization and accessibility of much cultural heritage [61], encouraging digital heritage and digital humanities to meet [62].

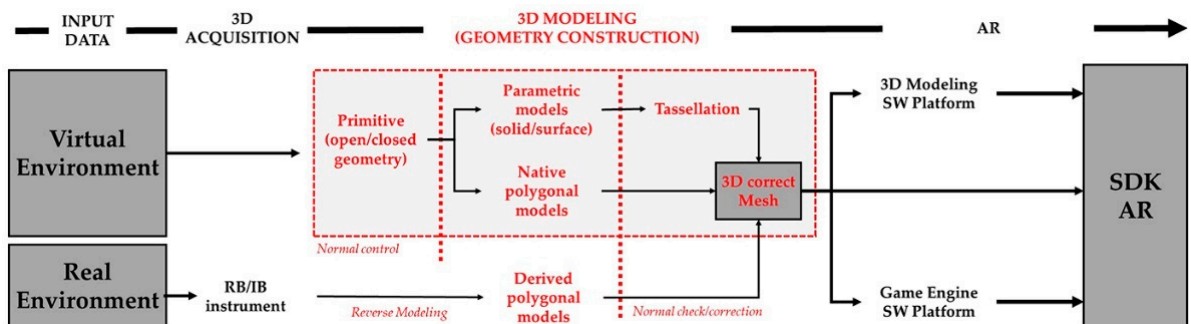

**Figure 9.** Diagram of 3D digital content creation for AR application. The 3D modeling passages and surface normal control are highlighted in red.

In the early 2000s, range-based systems were very limited in terms of sampling volume and scanning time, limiting their application largely to sculptural elements inside survey projects set out with topographic or photogrammetric techniques [63–66]. In 2005, the introduction of CW-AM systems led to a drastic reduction in the survey time,

preserving a similar resolution, accuracy, and uncertainty [67], and allowing the first complex 3D acquisition. In the past decade, the advancements in the scanning domain have been concentrated mainly on the reduction of equipment dimensions. Moreover, new methodological approaches have experimented with a multiresolution integrated pipeline [68].and the generation of correct, reality-based models [69,70].

In the passive system domain, the use of ever more advanced and cheap digital cameras has been accompanied by the evolution of software for image analysis, supporting photogrammetric processing, and the construction of 3D models [71]. The introduction of these automatic or semi-automatic processes, such as Structure from Motion (SfM) in the computer vision field [72,73], has gained these photogrammetric applications for the architectural survey and relative comparison works [74–76]. This trend, amplified in recent years by the increasing use of RPASs equipped with digital cameras [77], has supported the drive to build reality-based models referring to complex scenarios [78].

Today, the massive integration of active and passive systems makes it possible to obtain ever more accurate and complete surveying campaigns, ensuring redundancy data for integration and validation activity, heading definitively towards a multiresolution approach. Triangular polygons characterize reality-based models. The surface normals are calculated during the meshing step, matching the relative RGB data with the vertices or the polygons. The final 3D model's reliability is dependent on accuracy and uncertainty concerning the acquisition instrument and the reverse modeling process applied [79]. In the end, the point clouds or polygonal models can be manually or automatically classified [80], depending on the ontologies present in the architecture detected and in the analysis process. This subdivision can be highlighted and managed separately during visualization in AR.

### 4.2. Virtual-Based 3D Models

This category belongs to models created following bibliographic sources, iconographic sources, or 2D/3D survey information. These external data are translated into mathematical surfaces concerning the design choices or source interpretation. The interpretative effort and the control of relative tolerances are planned according to the aim(s) of the modeling process. Examples of architecture can be considered layered structures of geometrical, material, physical, and structural information necessary to read and fully understand an architectural organism. For this reason, virtual-based models produced by survey data define the basis of more consistent applications. Several BIM-based platforms offer the possibility of binding this information directly to the acquired data. The advantages of using mathematical modeling for AR are essentially threefold: On the one hand, the optimization of the management of 3D data simplifies the work of the graphics processor for real-time rendering. On the other hand, the flexibility in shape modification is helpful in the design process, and in the creation of diachronic built models. Finally, there is the possibility of better managing the stratification of information on different layers [59]. The models (Figure 9) are essentially divided into solid and surface models [81]. The former is semantically coherent with the architectural modeling, using the volumes and material consistency of the artifact. The material stratification favors constructive validation, and all types of simulations, but presents several bottlenecks in free-form surface generation and model management. All BIM-based modeling is part of this domain [82]. On the other hand, surface modeling proposes constructing building skins, improving computational management, and reproducing complex shapes. Moreover, it provides an abstraction of the real model, reducing it only to its outer shell. Mathematical and numerical models (meshes)—with some derivations, such as Sub-D modeling—belong to this domain. In general, the type of virtual-based modeling depends on the model's precision and the reconstruction aim.

### 4.3. Extension of 3D Models to AR

Reality-based or virtual-based 3D models may require an additional step to be translated into data suitable for AR (Figure 9). The more straightforward passage regards

direct uploading to an AR application, in which the automatic process of model adaptation is carried out. This case happens when a 3D model is ready for AR visualization without requiring post-production. Moreover, some modeling platforms foresee specific modules/plugins supporting editing process, i.e., AR-media (3DS, Cinema4D); Kubity (Google SketchUp), Mindesk (Rhinoceros), Marui (Maya), and Revit Live (Revit). The third, and more complex, solution regards the necessity of managing geometric, material, illumination, or animation modifications. In this situation, the modeling activity diverges from AR data preparation, adopting game engine platforms. These programs support complex 3D model management, interoperability of file formats, rendering, animation, and data interaction. On the one hand, they are limited by the complexity of the platforms—a significant obstacle for inexperienced programmers. On the other hand, the passage between different platforms requires verification of the transition of the 3D model's geometric and material characteristics. For example, in model generation, particular attention must be paid to the creation of shapes. The use of solid or mesh models, for different reasons, allows better directional control of the surface normals—a fundamental feature for the correct visualization of the model in real-time rendering.

In the mathematical surfaces, different directions of generative curves and main lines imply different distributions of normals. This aspect must necessarily be controlled a priori to reduce any subsequent post-processing work [18]. The most popular game engines are Unity 3D (Unity Reflect in the architecture domain), OpenSceneGraph, Unreal Engine, and CryEngine. Unity 3D is perhaps the most accessible platform that supports fully cross-platform development, but it does not allow real-time modeling. OpenSceneGraph is widely used for visual simulation, modeling, games, and mobile applications [83], but it is more oriented towards desktop and web-based applications. Unreal Engine 4 includes outstanding graphics features and is arguably the best tool for achieving accurate results. It enables the development and deployment of projects across all platforms and devices. CryEngine requires experienced developers.

## 5. AR System

Once the digital content is defined, the next step consists of defining the AR system that allows the user's engagement. This passage depends on some fundamental aspects (Figure 10):

- Devices: the HW system through which AR is used, analyzing instrument capacities concerning AR aims;
- Tracking and registration: related to the device, real environment, and virtual model reference systems;
- Interaction interfaces: relation between the virtual/real data and the user, crucial to visualization of AR use.

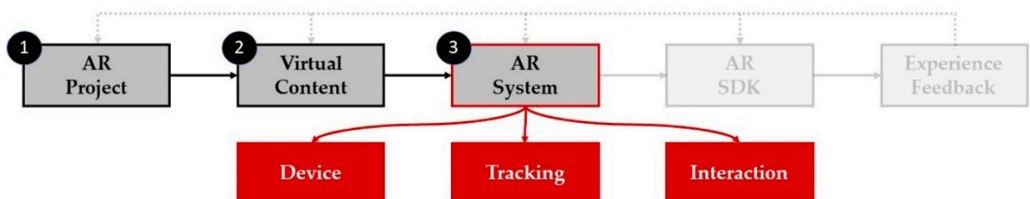

**Figure 10.** Block diagram of the main features of the virtual content creation.

### 5.1. Devices

The device is the main instrument in the AR process [3]; its characteristics affect the quality of the result, so it should be carefully evaluated during the design phases. Hereafter, the main device components in the AR domain are summarized below, in an order that reflects neither importance nor role, since each part contributes to the device's definition. Displays are the physical place where AR is represented and experienced through the overlap between real and digital data. Cameras and tracking systems are

responsible for acquiring reality and orienting the vision system concerning the real and virtual environments. The capacity of the computer device strongly influences these first two elements, responsible for processing the sensor's information and translating it into layered vision. In addition, two external elements may condition device use: on the one hand, connectivity, and the possibility of accessing digital content; and on the other hand, the use of external inputs as possible support for the device. Finally, social acceptance is treated as device use and users' experience, since it affects some display diffusion but provides valuable indications for future design.

### 5.1.1. Display

The display classification depends on the technology, the digital content, and the supported sensory channel [39,84]. Most of the applications related to architecture are addressed to the optical channel (Figure 11). The first type is the head-mounted display (HMD), a single-user device based on a video or optical operation [85]. The video-based HMD, called VST-HMD (video see-through), calculates the cameras' images and augments the acquired scene with virtual information, creating mixed images. This process requires a high data computation, but the result is very faithful and less tied to latency time [86]. The single-user mode can be applied to a collaborative scenario using projection screens or caves, much more commonly used in VR. The optical-based HMD, named OST-HMD (optical see-through), allows the user to view the real scene directly while virtual content is overlaid on the glass. This kind of display must present a low latency in the overlap between virtual and real data.

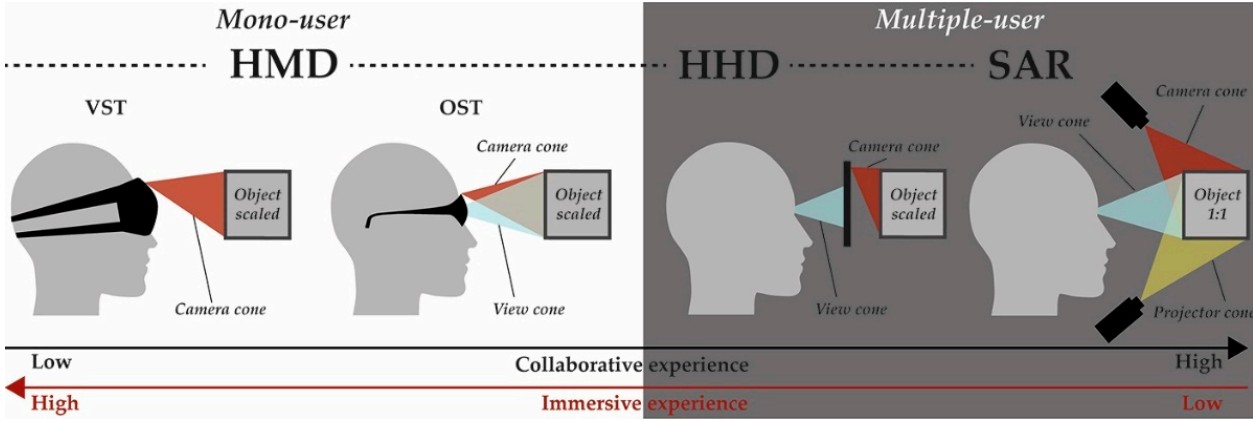

**Figure 11.** Functional schema of the different AR displays.

The second type of display is the handheld device (HHD), which combines a digital camera, inertial sensors, GPS, and display. This type of device is based on a VST approach by overlaying a camera-mediated view of reality with digital information. Most AR domain applications rely on these vision tools [20,43,45], which allow a collaborative approach and application in particular contexts, such as underwater [87].

A third type is spatial AR (SAR), based on the layering of virtual information on the real environment [3,88,89] using cameras, video projectors, or holography, basing their operation only on low latency, markerless tracking. This is a collaborative visualization with external devices and a direct in situ interaction at a 1:1 scale. Recent technological developments have allowed a remarkable development in the field of CH [90], or examples in the archaeological field [91] or BIM and collaborative design [92]. A critical evaluation of the whole system concerns the different immersive and collaborative display capacities. The wearable systems allow a low group experience but a higher immersion in mixed reality. For this reason, some projects—such as HySAR [93]—integrate multiple displays, obtaining the best performances from each instrument.

### 5.1.2. Cameras and Tracking Devices

Digital cameras are applied extensively to the architecture and cultural heritage domain, showing multiple applications [94]. There are various types of research regarding their acquisition performances, mainly referring to mobile phones [95], analyzing optical/geometric capacities [96]. Tracking systems are treated extensively in [1,49], focusing more on markerless systems in [97]. The device cameras are used for the reality–virtuality superimposition and the tracking of markers or features. Sometimes it may be necessary to combine cameras with external tracking devices if a hybrid approach is applied—for example, adopting electromagnetic, acoustic, and inertial sensors. An application such as AR Teleport by [45] leverages inertial sensors and a camera to track position. Integration with different sensors allows for enriched engagement and interaction with virtual models.

### 5.1.3. Computer

The processing unit responsible for merging real and virtual data is essential in the AR system, affecting the level of reliability of human vision. In this domain, there is an essential distinction between AR and MAR (mobile augmented reality): the former refers to all fixed PCs, while the latter devices (i.e., laptops, mobile phones, tablets, HMDs, glasses) are portable systems with the capacity to process data, and with integrated sensors that can be moved with respect to the real environment.

MAR devices are those most used in AR [98], and present specific pros and cons in their application [99]. The multiple cameras, inertial tracking system, and GPS make them particularly useful for outdoor AR applications. Indoors, instead, they rely on external anchors (marker or markerless) or positioning systems that exploit simultaneous localization and mapping (SLAM). This latter is an algorithm that maps the environment in which the user is located and tracks all movements [100]. AR apps containing this feature can recall physical and virtual objects within an environment according to users' movements. The main advantage of this technology is the ability to be used indoors, while GPS is only available outdoors. The project GEUINF [101] shows how to apply a markerless SLAM MR approach supported by the ARCore library to analyze indoor architecture facilities.

A fundamental role of the devices is represented by their ability to superimpose real and digital data in real time. The increase in computing capacity, well described by Moore's law, has affected AR and MAR systems, especially the latter. Significant efforts to improve runtime performance for mobile applications have been carried out in the past decade, speeding up hardware, software, and rendering improvements. Mobile AR systems are certainly among the primary resources in AR prospects, developing some important AR components, such as multicore CPU parallelism, GPU computing, cache efficiency, memory bandwidth saving, rendering improvement, and computation outsourcing. The use of devices is closely linked to the capacity to access the data network.

### 5.1.4. Network Connectivity

This topic is crucial for MAR systems. There are two different aspects to evaluate, intertwined among them: the type of network, and the data transmission capacity [99]. Regarding the former, MAR can work with local-based or stream-based applications. In the first case, the digital model's database is downloaded to the same AR device, suggesting a self-consistent system. The database is disconnected from the AR device in the second case, so it is necessary to stream digital data via Wi-Fi, Bluetooth, or mobile network. In general, MARs present different network interfaces for connecting directly with a remote server through a mobile network or Wi-Fi. Moreover, wearable devices usually do not have a network interface, but have Bluetooth connections with an external device. The data rate is the consequence of the connection type and the distance from the routers/antennas. As for mobile networks, there are great expectations with the advent of the 5G system [102]. Two quick concepts descend from this topic: On the one hand, mobile network systems are currently favored for outdoor applications, and 5G predictions suggest that they will also

boost indoor applications. On the other hand, the type of connection heavily conditions the type of information conveyed, the tracking, and the real environment's recording activities. Digital data such as videos and 3D models in markerless tracking require a consistent data stream, with the risk of increasing the latency time and the correct AR perception.

### 5.1.5. Input Devices

AR input devices allow user interaction to be moved from the graphical user interface present in the device (GUI) to intuitive and natural systems based on gestures or sound/visual input. The devices used are usually wearable (i.e., gloves, bracelets), and closely related to the domain of the application and the purpose of AR [4]. If mobile devices are used, it is possible to foresee integration between the touchscreen, the microphone, and the tracking sensors. An example in the architecture domain is offered by the Tooteko AR application [103], which uses near-field communication (NFC) sensors connected to a 3D prototyped model of an artifact as an input device, returning audio content when the user touches the model. The integration between haptic systems for MAR is described in [104]. Pham and Stuerzlinger propose comparing the different controllers, focusing on the flexibility in using the 3D pen for VR and AR applications [105]. Electronic skins have recently been investigated as one of the most promising device solutions for future VR/AR devices [106], while the ARtention project [107] illustrates how input information can come directly from retinal motion. Finally, the MARVIS project proposes integrating HMDs and mobile devices for optimized data management in augmented reality [108].

### 5.1.6. Social Acceptance

Some elements—such as device intrusion, privacy, and security—influence AR systems' social acceptance and diffusion [99]. Device intrusion is perceived as a device's physical presence and visibility within everyday life, affecting both the aesthetic and functional aspects. The miniaturization of display devices and an increasingly interactive and natural interface may reduce this factor. Early applications with backpack laptops and HMDs induced significant device intrusion. Device size reduction and performance advances have partially mitigated this problem, leaving some areas for improvement. On the one hand, the aesthetic design of some smart glasses (i.e., Microsoft HoloLens, Google Glass) remains quite invasive, and not very "similar" to standard glasses. On the other hand, gesture optimization reduces the pointing activity of cameras and physical user discomfort [109].

The privacy problem is felt in both the private and public spheres. The data ownership of recorded images, videos, and sounds saved in the cloud is mainly related to social networks. The use of an OST-HMD that registers information in space introduces a severe privacy issue. For example, the movement "Stop the Cyborgs" has tried to convince people to ban Google Glass in their premises, while many companies have banned Google Glass in their offices.

Finally, the last aspect is related to caution. The use of visors that superimpose additional information over the real world can expand the knowledge of the environment. However, the presence of "external" elements in the scene catalyzes the gaze, reducing attention to the real world. This translates into a safety problem when the device is in motion, especially in vehicles. Hence, the design of the positioning of the supplementary information on the visor concerning the real scene becomes crucial to preserve attention. These aspects take on a different role in the educational domain; there, the acceptance is based on what users expect from this technology. A central role must balance the technological aspect with the pedagogical ones, offering AR applications that guarantee continuous engagement during the interaction, self-learning capacity, and parental involvement. It is also important to keep in mind the context or educational conditions of the students and the AR platform's typology [110].

## 5.2. Tracking and Registration

Tracking means following the user's point of view concerning a scene—real or virtual. In AR, this activity primarily aims to acquire data relevant to superimpose the user's view with the virtual one, positioning the virtual model in real space. The tracking registration, carried out mainly thanks to the camera, is essential for the correct real-time operation of AR. The precision and speed of acquisition fix the update rate and the superposition accuracy of the virtual model on the real one. The first step, defined as calibration, defines the orientation system in which the camera records the first scene. After this "time zero", the following steps are referred to as tracking activity, updating the initial reference system according to the system changes, and calculating the roto-translation matrix between the camera center and the real scene's position. This operation allows the correct alignment of the real and virtual models, called registration [1,111].

The elements in the scene that enable tracking can be artificial or natural. Artificial refers to all those elements explicitly created to support the camera in recognizing the scene. These include markers (standard or infrared) and tracking sensors external to the camera (electromagnetic or hybrid tracking). Natural elements are elements already present in the scene, whose shapes are recognized and used for tracking. In this case, therefore, no additional elements are needed, and we talk about markerless tracking.

The automatic recognition of targets or shapes in space is strongly related to the point of view. Extreme perspective distortion of the subject through the device can cause failure in recognition. This limitation happens more frequently for flat markers than in recognition of 3D shapes. The assumption of an inconsistent point of view may cause the failure of the recognition process or a slowdown in processing, in case the camera moves quickly. For this reason, the choice of targets and their position in space must be carefully evaluated in the design phase [41,42,112], as well as the tracking system, still suggesting ubiquitous hybrid tracking systems [113].

The data fusion between different sensors allows increasing the device's capacity for scene recognition, solving specific bottlenecks in scene recognition at a large scale. Some problems arise if the GPS coverage is low, so tracking by location is not suitable, or if the geometric characteristics of the artifact do not allow easy recognition. A primary solution can be the hybrid image recognition technique, merging image-based markers referring to the monument pictures and QR codes [114]. Reitmayr et al. [115] suggest integrating an edge-based tracker with gyroscopic measurements and a back store of online reference frame selection to re-initialize automatically after dynamic occlusions or failures. Some tracking systems enlarge users' participation, activating co-design actions. In this sense, the Urban CoBuilder project shows how citizens can collaborate on urban environment projects using AR. The use of multiple markers and smartphone gyroscope sensors activates co-design strategies at an urban scale [116]. Some devices are based on a markerless approach, facilitating the outdoor experience. For example, the HMD HoloLens (Microsoft) has inbuilt processing units to handle all computational needs, offering a toolkit compatible with the Unity game engine. The user experience offered by HoloLens pushes a large-scale visualization, suggesting an immersive and natural feel of data visualization [117]. This capability also facilitates cultural heritage experiences, merging all the resources required for tracking, computation, and displaying virtual objects and audiovisual elements in the device [90]. The markerless approach can also be helpful in moving from urban to architectural and detail scales, i.e., suggesting a guidance system with the AR device and camera sensor to display the virtual model in a designated position in the real world, facilitating the assembly process [118].

### 5.2.1. Marker Tracking

Image recognition algorithms automatically recognize AR markers in the scene [119]. Some of these fiducial markers (Figure 12), such as QR codes (quick response codes) and Maxi codes since the late 90s, are commonly considered to be two-dimensional cryptograms or barcodes that redirect to other databases and programs, typically online, activating

multimedia information or applications. In the last decade, they have also been used in AR [120]. An AR code is usually a QR code with a marker that redirects the user to the AR web app or mobile app.

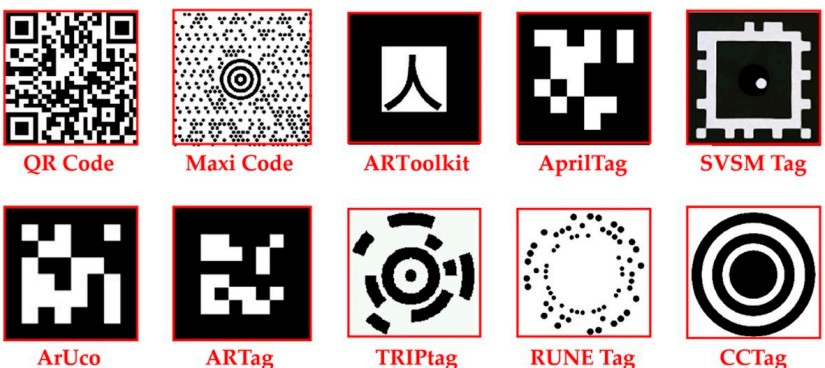

**Figure 12.** Some examples of markers recognizable through image recognition.

The markers can be passive—printed on any surface—or active, defined by infrared emission. In both cases, the calibration and the tracking activities occur with cameras that work on the visible or infrared spectra. Most AR applications use the first type, while the second is helpful in AR indoors, where the non-ideal lighting conditions can cause greater complexity in recognizing passive markers.

The use of AR markers in architecture can depend on several factors, first of all:

- The relationship established between the marker and the real scene;
- The possibility of preserving its location;
- The lighting conditions.

Regarding the first point, it is not always possible to directly place the marker on the artifact to preserve the surface without altering the original aspect. In this sense, using totems with markers next to the architecture can solve the problem, planning their spatial position concerning the building and urban design simulation carefully [121]. Moreover, the second aspect concerns the conservation of the marker in terms of preservation of position and support. This limitation is particularly evident outdoors, where atmospheric agents and less scene control can make its application very difficult. The factor presents a minor impact in indoor environments, mainly if there is an active surveillance system; this is the reason why markers are often used in museum contexts [122]. Finally, lighting conditions strongly influence the target recognition activity. Although this problem is more limited in outdoor conditions, it can become significant in indoor ones. Active or passive IR markers can be used in such situations [123]. In the case of active IR markers, diodes that periodically emit IR light are used. Moreover, passive markers consist of retroreflective materials that reflect incoming IR light to the source. Multiple cameras frame a scene illuminated with IR light, paying attention to preserving the line of sight between the IR source and the reflector, in the meantime avoiding ambient IR radiation that may disturb this reception. In general, there is no relation between marker recognition and network data access, since the process depends only on the device's camera. Therefore, lack of network access is not binding, as AR content can be uploaded either in an external online server or directly on the offline device's local application.

Another tracking possibility is based on sensors outside the camera. These can be electromagnetic, inertial, acoustic, or hybrid. The former bases its measurements on the intensity of the magnetic field existing between a base station and a measurement point, according to different directions and orientations; it has low latency and high responsiveness, but the measurement is subject to interference from other magnetic fields near the tracking space, so it requires a controlled environment. The inertial tracking system uses gyroscopes and accelerometers to measure the rotation and motion of a given target, allowing the calculation of position and speed. This method allows for a high

update rate with low latency and low cost but suffers from the accumulation of minor positioning errors made by the gyroscope and accelerometer, so it is often integrated with other tracking systems. On the other hand, acoustic tracking estimates the position of a viewpoint by calculating the time it takes for ultrasound to travel from a target (emitter) to a sensor, which is usually kept fixed in spatial tracking. These systems suffer from low signal update rates caused by the low speed of sound and ambient noise that causes measurement errors; thus, integration with other tracking methods is also advisable in their case. Finally, hybrid tracking consists of the fusion of the tracking methods described above to obtain better results than using them separately.

Among the integration between tracking systems, a first example is between inertial tracking with IR markers and other tracking methods, such as GPS and camera [56]. This latter solution is suitable when POIs are very close to one another, ensuring a more flexible and reliable device recognition [124]. Other applications in the CH domain using hybrid tracking are described in [40], while the i-Tracker system, optimized for in situ analysis, merges depth-sensing cameras and inertial measurement unit (IMU) sensors [125].

### 5.2.2. Markerless Tracking

According to the survey, the markerless approach is based on the camera's orientation and automatic recognition of some geometric features of the real environment. Compared to the marker approach, this method can allow real-time tracking, but the large amount of data to be processed can slow down the recognition process. [37]. In this case, concerning the three main constraints shown in the use of markers, only the ambient lighting conditions affect feature recognition. Markerless tracking is based on the recognition of specific geometric features (feature-based), such as corners and edges of buildings, or shapes (model-based) within the scene [126]. The concept of a feature-based approach is finding a correspondence between 2D image features and their 3D coordinates in the scene. Recent approaches show the possibility of using a structure from motion (SfM) methodology to generate 3DCG (three-dimensional computer graphics) model objects on a live video feed of the surrounding landscape acquired via a camera [127].

On the other hand, model-based methods use CAD models or 2D drawings that outline the main objects by identifying the lines, edges, or shapes present in the model. This method is applicable in indoor and outdoor environments, but the observation point intensely conditions it. In this sense, identifying planes in the environment and applying 3D to 2D feature projection allows the estimation of the camera position without any SDK system [128]. Some recent experiments have shown that it is possible to merge depth information given by a sensor with the position of a camera, establishing a relationship between the 2D space of the image and the 3D space of the scene, resulting in a reliable estimate of the camera position [56]. The markerless approach is much more flexible and intuitive, applicable both indoors and outdoors as long as the image database of the scene is connected to the location, but suffers from processing times that often lead to a delay in data registration.

### 5.3. Interface Interaction

The interaction between users, real scenes, and simulated data is a third crucial aspect of the AR system. In this domain, the main research aim is to build intuitive and natural interfaces. Many kinds of research have been carried out in augmented reality over the past 20 years in different specialized areas, regarding tangible user interfaces (TUIs) and human–computer interfaces (HCIs) [4,40,45,129,130]. Interaction in AR has a crucial impact on the sense of virtual information in the user's reality, working on the visual perception of the non-physical world superimposed on the real world. This is a crucial aspect in the educational domain, in which the interface interaction design plays a central role in knowledge transmission and engagement [131]. There are four types of interfaces closely related to the AR domain: tangible, sensor-based, collaborative, and hybrid interfaces (Table 1).

**Table 1.** Interface comparison concerning AR experience.

| Interface | AR Experience | | |
| --- | --- | --- | --- |
| | Content Use | On-Site | Remote |
| Tangible | Personal | YES | NO |
| Sensor-Based | Personal | YES | NO |
| Collaborative | Group | YES | YES |
| Hybrid | Personal/Group | YES | YES |

### 5.3.1. Tangible Interfaces

A tangible interface allows direct interaction by manipulating physical objects, combining computer-generated content with physical environments in AR [132]. In this way, the physical object acquires the dual role of representation and medium of the interaction process. However, it is essential to distinguish whether the physical object interacts with the virtual information or is augmented through a relationship between AR and TUI. The MAR application makes it possible to apply an intuitive tangible object as a tangible user interface, manipulating the digital AR content [133]. In general, tangible objects such as touchscreens define a relationship between the real and virtual worlds—a common practice in architecture and cultural heritage [134]. On the other hand, the possibility of touching and handling some ancient artifacts is often precluded for conservation reasons. Different is the possibility of touching replicas of artifacts, as in the Tooteko AR application [103], whose contact becomes the tool for accessing digital content

### 5.3.2. Sensor-Based Interfaces

These types of interfaces use sensing devices to understand and capture natural modes of interaction. They do not detect user input but define an active perception of the system towards specific signals appropriately encoded and read by the system. Standard sensors include motion detection [135], gaze tracking [107], and speech recognition for visual content [136]. Most of these sensors have little presence in the AR world. Recently, in the MAR domain, more and more methodologies use sensor-based approaches to analyze virtual space, handling semantically classified point clouds or models [137].

### 5.3.3. Collaborative Interfaces

The collaborative interfaces are based on the integrated use of multiple displays and devices, such as HMD and SAR, sharing activities and contents remotely or on-site [3]. In the case of shared visualization in presence, the use of helmets is more frequent, allowing each person to start a personal experience of the virtual model. In remote conditions, common virtual space is used, displaying the same content within it. The research topic of collaborative interfaces in AR is strongly related to the world of robotics. A state of the art that highlights the intersections between AR and computer-supported collaborative work and related future research scenarios can be found in [138]. Some of these applications are dedicated to the navigation of architecture, interacting with other users who experience the same simulation [139]. In the AEC realm, collaboration on building monitoring and verification is increasingly essential, as highlighted in [140].

### 5.3.4. Hybrid Interfaces

Hybrid interfaces offer the same possibilities as multi-user collaborative interfaces [3,10], with a different purpose: the former also allows single-use AR, whereas the latter cannot. The users use a glove, voice commands, and a sensitive multitouch surface to multimodally interact with the system and collaborate to navigate, search, and display data. Framed in the urban scale representation, this can be understood as a space exploration tool for graphical visualization and communication [141]. The MARVIS project provides other examples of the integration between HMDs and mobile devices for data analysis [108]. The integration between computers and HoloLens devices for reading and understanding

three-dimensional data is offered in [142]. In the end, [87] shows how this approach can be helpful to overcome multimedia conditions.

## 6. AR Development Tools

Augmented reality software development kits (AR-SDKs) have a "bridge" function between the device and the virtual content through a platform devoted to data integration, management, and visualization. Each one offers a specific GUI (graphical user interface) to connect the user's virtual content. These are intended as "interaction metaphors" between the input and output data, highlighting the communicative capacities and purposes of AR digital content in the real world. Several parameters define the use of these AR-SDKs, and their possible application in the architecture domain: cost, supported platform, image or target recognition, 3D and tracking, support by game engines, cloud or local databases, geolocation, and SLAM [143]. Moreover, in this paragraph, a different clustering of AR-SDKs based on accessibility and availability is proposed, dividing them into open-source, free (proprietary), and paid (commercial) (Figure 13). The other specific AR-SDK characteristics will be evaluated accordingly. The open-source systems guarantee their implementation through access to the source code. In contrast, the others differ only in terms of being free or paid, but do not allow a system modification.

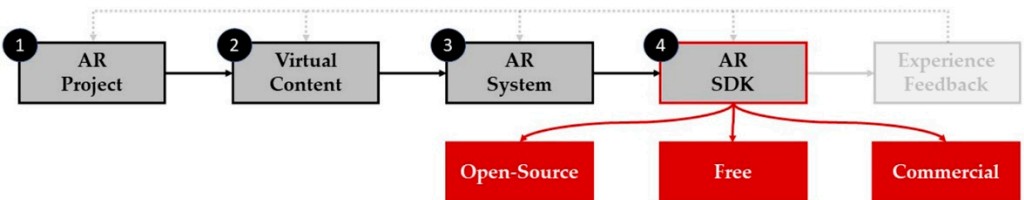

**Figure 13.** Block diagram of the main features of the AR-SDKs.

Most of the leading AR-SDKs were developed between 2007 and 2012, but the app scenario has profoundly changed after 10 years. The number of platforms available on the market has increased significantly. Here, the main multipurpose platforms, or those devoted to the architecture domain, are discussed (Table 2).

**Table 2.** List of SDKs and relative characteristics.

| SDK | Typology | Platform | Tracking | Domain |
|---|---|---|---|---|
| ARToolkit | GPL [1] | Multiplatform | Marker | Generic |
| DroidAR | GPL | Android | Location/Marker | Generic |
| AR.js | GPL | Multiplatform | Location/Marker | Generic |
| EasyAR Sense | GPL/Commercial | Android | Marker/Markerless | Generic |
| Apple ARKit | Free/Proprietary | iOS | Location/Markerless | Generic |
| Google ARCore | Free/Proprietary | Android | Location/Markerless | Generic |
| ARloopa | Free/Proprietary | Multiplatform | Location/Marker/Markerless | Graphic |
| Archi-Lens | Free/Proprietary | Multiplatform | Marker | AEC/Design |
| Layar | Commercial | Multiplatform | Location/Marker/Markerless | Generic |
| Wikytude | Free/Commercial | Multiplatform | Location/Marker/Markerless | Generic |
| Vuforia | Free/Commercial | Multiplatform | Location/Marker/Markerless | Generic |
| MAXST | Commercial | Multiplatform | Marker/Markerless | Generic |
| AkulAR | Free/Commercial | Multiplatform | Location/Marker | Architecture |
| Augment | Commercial | Multiplatform | Marker/Markerless | eCommerce/AEC |
| ARki | Free/Commercial | IOS (Android soon) | Location/Markerless | AEC |
| Fuzor | Commercial | Windows | Markerless | AEC |
| GammaAR | Commercial | Multiplatform | Markerless | AEC |
| Dalux TwinBIM | Commercial | Multiplatform | Markerless | AEC |
| Fologram | Commercial | Multiplatform | Marker | Generic |

[1] General public license.

### 6.1. Open-Source SDKs

The main open-source and multipurpose SDKs are ArRToolkit, DroidAR, AR.js, and EasyAR (Table 2). ARToolKit is a vision-based AR library that includes camera position/orientation tracking, camera calibration code, and cross-platform development. It uses video tracking capabilities that calculate camera position and orientation relative to square physical markers or natural feature markers in real time, solving both viewpoint tracking and virtual object interaction. Distributed with complete source code and initially used for PC applications [144], it was the first mobile AR-SDK, seen running first on Symbian in 2005. It was tested in many applications for architecture and design purposes [24], to support collaborative design decision meetings [145].

DroidAR was developed in 2010 and was designed to create AR applications for the Android operating system, both for location and image recognition. It is the only open-source (GPLv3) AR SDK dedicated to Android applications, supporting build location-based and marker-based AR experiences. Evidence of this tool appears in [14], and environmental documentation in [146].

AR.js is an effective, JavaScript-powered, open-source (MIT license) augmented reality SDK for the web. This solution enables AR experiences on the browser without having to download and install any app. AR.js runs are very fast, reaching 60fps, and can be used on any mobile platform device. A recent example is represented by the METEORA project applied to built heritage [147] and learning object creation for geometrical space analysis [148].

EasyAR Sense, previously known as EasyAR SDK, is a standalone SDK that brings new algorithm components and platform support to increase AR capabilities. It supports sparse and dense spatial maps, as well as motion, surface, 3D object, and planar tracking.

### 6.2. Free/Proprietary SDKs

In early 2018, Google launched ARCore, an open-source (Apache 2.0 license) augmented reality SDK for bringing compelling AR experiences specifically to Android devices. ARCore employs three main capabilities: motion tracking, environmental understanding, and light estimation. ARCore's motion tracking technology uses the phone's camera to identify points of interest and tracks how these points move over time. In addition, ARCore can detect flat surfaces and estimate the average lighting in the surrounding area. The LayART project suggests the generation of 2D layouts from indoor images using SLAM and ARCore [149], while Palma et al. used this application to superimpose real and reconstructed vault systems [150].

As with the ARCore system, it is essential to cite ARKit, a system related only to the IOS platform, which integrates depth sensor information with depth API, location anchoring, and facial tracking. The merging of scene geometry calculation, person occlusion analysis, instant AR with LIDAR plane detection, and motion capture constitutes a powerful AR system. After almost a decade of using the free software Aurasma, today, a possible alternative with some implementation is offered by ARloopa—an augmented reality (AR) and virtual reality (VR) app and game development system providing advanced AR and VR services: cloud-based AR, custom-branded AR app and game development, virtual reality app and game development, and 2D and 3D content creation. Its simplicity of execution, while limiting functions, can help expand AR applications. An appealing and promising project concerning AR in the architecture domain is Archi-Lens, in which blockchain is combined with AR to store documents and AR models of the construction plans, as well as using AR to design and keep track of the state of the construction site [151].

### 6.3. Commercial SDK

The most used and best known multipurpose commercial SDKs, not devoted to the architecture domain alone, are Layar, Wikitude, and Vuforia (Table 2). Layar is the most widely used AR-SDK for service localization; it can store POIs in a remote database (DB) and retrieve the associated information based on the user's position, using the Blippar

augmented reality SDK to enable quick and easy mobile AR development. These features make this kind of application ideal for outdoor wayfinding applications and experiences in architecture and urban planning [5] and create a framework for architecture and building engineering education [152].

Wikitude is a platform released in 2008 that leverages both location-based and vision-based tracking. For the latter, it supports images and 3D model recognition. The generation of 3D-model-based targets is currently in beta version (SDK 7), but from 2017 it introduced SLAM technology and instant tracking, working with three-dimensional structures and objects. An interesting application for partially damaged or destroyed buildings to help visitors interact with the city's monuments is offered by [49], using this geo-localization to interact with archaeological sites [153].

After removing Junaio (Metaio) from the market, Vuforia has become the most widely used toolkit for most developers. Its simple integration with Unity3D allows the visualization of refined 3D models, and a quick and easy cross-platform development. It supports various 2D and 3D target types, including image targets, 3D multitarget configurations, and a form of addressable fiducial marker known as VuMark. The author of [154] presents a framework for visualizing 3D artifacts with Vuforia in archaeology, while [155] explains the systematic structure, recognition of the target library, and the working process of the virtual exhibition hall system. At the urban/landscape scale, [156] suggests an application, while [157] proposes the visualization of interpretative 3D models, and [158] proposes an AR BIM-based application.

Furthermore, some additional applications are implemented in the architecture domain. One is Maxst, a powerful AR-SDK system that combines virtually enhanced physical reality with AR (augmented reality), AI (artificial intelligence), and computer vision technology. MAXST provides various solutions, supported by VPS (visual positioning service), to define the user's location by constructing a 3D spatial map for both in/outdoor AR content creation, instants, objects, QR codes, images, and marker trackers, VisualSLAM, and Cloud Recognizer. An application related to interior design and planning architecture furniture can be found in [159]. Moreover, there are several AR-SDKs designed to support architecture design and visualization. One strictly connected to Autodesk is AkulAR, allowing the user to experience 3D digital models at real size, geo-located in the real world, using only a smartphone or tablet. IN2AR is a non-new ActionScript3 and Adobe Native Extension Library or Unity3D plugin that detects images and estimates their pose using standard webcams/cameras or mobile cameras. The pose information can place 3D objects and videos onto the image for AR applications. Augment supports the AEC domain with easy configuration; it is based on the Vuforia augmented reality SDK and OpenGL. Meanwhile ARki is a real-time AR visualization service for architectural models, which incorporates AR technology into architecture by providing 3D models overlaid onto existing 2D floor plans, with multiple levels of interactivity for both design and presentation purposes; it includes real-time shadow analysis and material selection. Fuzor is a real-time rendering app integrated with Revit, allowing users to move around, visualize, annotate, and inspect BIM information, while also syncing changes between the two tools; it supports various measurements, clash analysis, lighting analysis, color and visibility filters, cross-section and section cut rendering, and walkthrough video rendering with BIM information embedded. GammaAR is an application that uses AR technology to overlay 3D models using smartphones or tablets for the AEC industry; it helps the users to compare the real work with the planning information contained in the project. Moreover, Gamma AR connects the construction site and the office via a portal, enabling the user to access any annotation, photos, issues, or reports at any time. Dalux, developed in 2005 as a BIM viewer application, allows users to access the BIM model and sheets using smartphones or tablets; moreover, it developed a new technology tool called TwinBIM, which merges the BIM model with the physical world; it is compatible with Google ARCore and iOS ARKit, and works by scanning the physical environment and creating a 3D map of the surroundings, knowing exactly where to place the BIM model, aligning and anchoring it to the physical

world. Fologram collects applications and plugins that run in mixed reality devices and integrate with Rhino and Grasshopper. In Fologram, geometry, layers, and grasshopper parameters can be modified in real time to create powerful tools for design, communication, collaboration, and fabrication using smart glasses or mobile devices. Finally, AR CHECK is an application that makes the use of 2D prints on construction sites unnecessary, and eliminates human error, preventing mistakes in the prefab construction industry.

## 7. AR User Experience

To conclude the AR development, its application must correspond to those design criteria planned in the design step, with feedback on the knowledge model. The AR application is a moment of augmented experiential deepening of context and content, affected by several factors (Figure 14). In the architectural domain, the complexity of the case study analyzed, the point of scene observation, and the level of interaction play crucial roles. Moreover, the user experience achievement must be coherent with the project aims; this can be verified through a feedback system, optional in some applications and mandatory in others (Figure 6). The intersection of context, content, and purpose may generate different punctual and localized experiences, or ones developed within precise knowledge pathways (Figure 15).

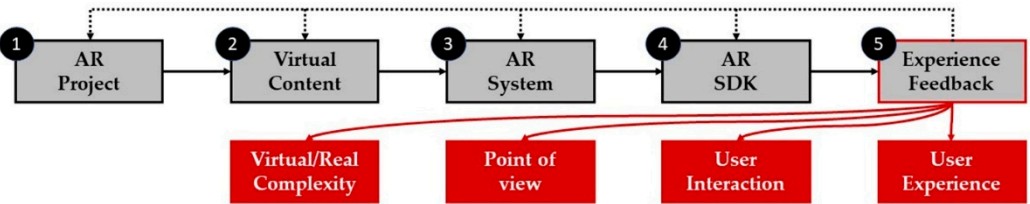

**Figure 14.** Block diagram of the main features of the AR feedback Experience.

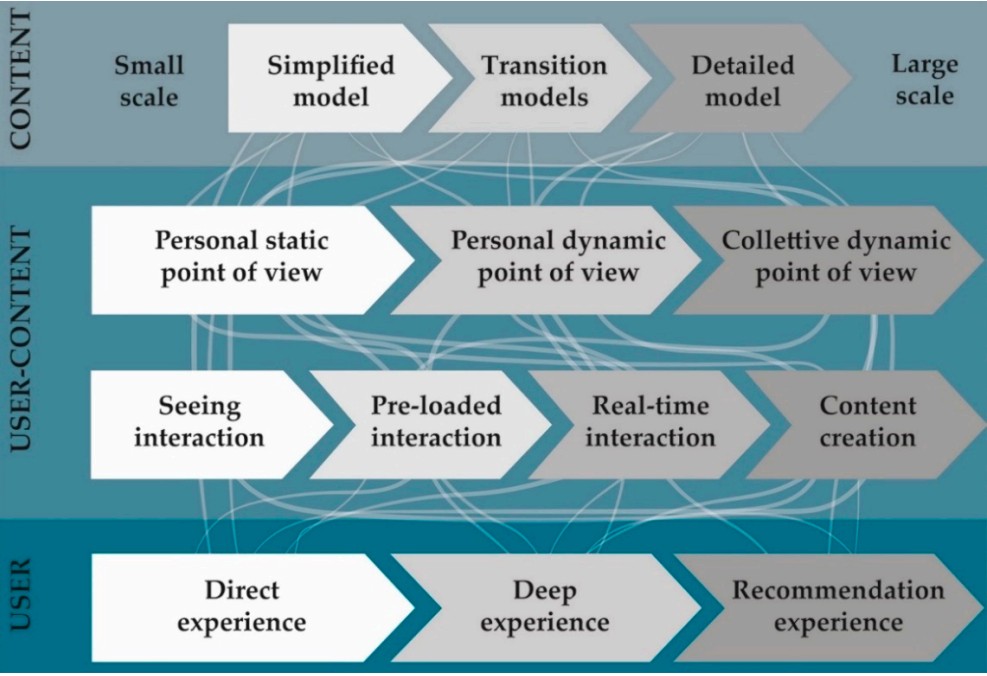

**Figure 15.** Block diagram showing the relationships between some critical aspects related to the AR user experience. The graphical network behind the pipelines outlines some possible interactions between all the elements.

The first point concerns the relationship between the model complexity and the capacity to transmit knowledge through AR. Architectural complexity is a vast topic,

ranging from the artifact's size to its geometric and material complexity, from the functional to the hierarchical relations of the individual components and their mutual inferences. Among these elements, the scale of visualization and information is crucial in the domain of AR. Architecture is characterized by a substantial variation of scale, from the territorial level to the individual construction detail. Each of these scales contains a different data level, which can be included and integrated into a multiscale 3D model. However, the experiential mode of AR, which usually provides fewer tools for visualization and management of digital data, reduces the layered model's capacity, limiting the investigation and interaction with different levels of scale. Moreover, the complexity of information related to the individual scales poses both a problem of managing massive digital data and usefulness in experiential terms. Abstraction activity, through geometrical simplification, may lead to a better and faster understanding of phenomena or spatial relationships. Therefore, it may turn out to be more functional when using sequences of models to increase complexity concerning the knowledge and purposes of AR in the AR domain.

The experiential model also refers to the point of view (Figure 15), central both for tracking and visualization activities. The dynamic relationship between the camera position and a 2D marker/image is vital to deepen the visualized element and its construction rules. This movement becomes necessary in 3D markers/artifacts to experience the real object and initiate many visualizations connected to different target areas. The introduction of the movement may involve both target and camera systems if the 2D marker is not bound to fixed elements; otherwise, it all depends on the camera reference system in the space. In both cases, the user's freedom to discover additional information from the point of view starts an individual and collective educative experience. First, the user can develop his knowledge model, investigate the relevant aspects of the subject more deeply, and start an autonomous learning path framed in structured educational models. This experience is intertwined with a collective one. Comparing the different individual experiential models on the same subject may improve critical knowledge by comparing the different experiences. In the end, particular attention should be paid to a non-immersed visualization of interior architecture. The low immersive level of AR, unless using HHD systems, provides few solutions in this case. One such solution consists of making the external envelope transparent, allowing an internal visualization. Otherwise, it is appropriate to integrate the visualization with a predefined system that gives access to spherical images, alternating an AR and a VR vision.

Another issue is the interaction with the virtual model (Figure 15). It is well known that the interaction of AR is much lower than that of MR and VR (Figure 3). Nevertheless, it is possible to identify some levels of interaction with the subject. The first is the low-level one, based on the overlap between virtual and real data. In this experience, the personal interaction consists of choosing the points of view to investigate and deepen the digital data, starting a critical path based on observation. An intermediate level of interaction [154] foresees some punctual functions on the display that start preloaded processes, such as variations in the visualization or animations. A third level concerns the real-time interaction of the user with the digital data, varying the visualization according to the user's actions. This increasing interaction is related to the value of the personal knowledge path in the architecture and the specific purpose of the AR. Moreover, the more profound link between interaction, experience personalization, and content concerns new content creation. This last level refers to a possible research domain devoted to authoring new personalized digital models superimposed on reality through user–content interaction. Some variables, such as recorded reality data, tracking shapes, or GUI boundary conditions, managed by an AI system, may suggest new shapes in the space, introducing an absolute personalization of the cognitive path.

The last aspect refers to the experience feedback system for the validation of the process (Figure 15), identifying the different improvement boundaries. Feedback occurs concerning the AR goals. First, feedback passes through the direct experience, and can be considered an added value compared to the available data in the actual scenario. In

this case, feedback can happen via quick choice responses to better address the target group's needs for the proposed cultural itinerary. The second level of feedback concerns an evaluation of the knowledge and understanding process:

- How this AR visualization transmits content;
- How this content is rooted in personal experience;
- How AR experience is translated into a better understanding of the real phenomena.

It is necessary to plan a deeper system to evaluate feedback as a new and more effective channel for learning. According to the specific case analyzed, this process can occur through interviews or questions to test this learning stream. This planned feedback system, appropriately weighted concerning the type of user, allows the system to be implemented. In the educational domain, the group experience can also be evaluated. The authors of [160] propose a mobile AR framework for peer assessment, in which students can submit their work and evaluate that of others. A final aspect explores the integration of augmented reality and data mining to record how a given application is used for a specific goal. The specific user–model interaction through a specific GUI can be recorded anonymously and processed by the system. Therefore, these usage data can be employed and implemented in a machine learning system to define new scenarios, optimize the system, and improve the recommendation abilities.

## 8. Applications

In the architecture domain, there are several experiences in more than 20 years of research on a global scale. In the final section, a synthetic and comparative overview of the possible AR applications in this domain is offered, supported by citations of some significant examples.

The subdivision of such a broad topic appears complex due to the multiple domain classifications. It was decided to group the application experiences into three macro-areas of research: AR for build deepening and enhancement, AR for the architectural design and building process, and AR for architectural training and education. The first area addresses historic and historicized architecture, which flows into cultural heritage, in which AR for architecture plays a fundamental role in knowledge, conservation, and visibility. The second area mainly focuses on AEC for new buildings, deepening AR in the interaction phase between designers and users (co-design) and managing the complex process of building construction. Finally, the last topic discusses knowledge transfer through didactic, educational, and training paths for cultural aims and professional training.

The selection of experiences refers to the following objective parameters, shared in the three areas, and not in order of importance: keyword, date publication, abstract feedback, and text analysis. The use of keywords makes it possible to narrow the scope of the search. Some of them have a cross-cutting value on the three domains (for example: architecture, built environment, augmented reality, state of the art), while others vary according to the domain in this way:

- Area 1: Cultural heritage communication/valorization, build enhancement/management, buildings storytelling, cultural tourism, monument exploration;
- Area 2: AEC, design process, project planning, co-design, monitoring construction, building process, BIM pipeline, collaborative process;
- Area 3: Education, training activities, knowledge transmission, EdTech, professional skills, civil engineering study, assembling learning.

According to the order of keyword coherence, the publication date mainly —but not exclusively—targeted two periods. The first addressed the first or "founding" experiments in the field, highlighted by search engines as those with more citations, more visibility, and more rooted in the field of interest. However, this process has also highlighted significant articles in the past 20 years. A second search started in the year 2021 to present an overview of research updated to recent years. These search criteria allowed the definition of the first

selection of articles. The selection process was then refined in the contents, going from the abstract to the paper, verifying data coherence and relevance.

### 8.1. AR for Build Deepening and Enhancement

There is a strict connection between understanding, knowledge, and reading of the existing environment with built enhancement. Indeed, this distinction is directly related to the different target groups to which the application of AR is addressed. Starting more broadly from the basis of the valorization of the built environment, outlined in the domains of both architecture and archaeology, there have been many applications and case studies analyzed in the last 20 years. The pioneering experiments in the early 2000s aimed to highlight some potential of AR in the reading of the archaeological heritage [161]. It is worth mentioning the ARCHEOGUIDE [40], ARICH, and ARCO projects [162]. Ten years on from these experiences, a critical AR diffusion [9] is highlighted, exploiting its relevance and adaptability, and showing the first MAR applications [43,163].

Based on many experiments [50,164–169], it is possible to state that there are three primary applications of AR experimentation and development for architecture and cultural heritage:

- Applications to improve the visitor experience in contexts in which heritage is placed, distributed, or collected. Cultural tourism is an essential economic and social contributor worldwide. This area is more related to education [170], valorization, and tourist routes in general, improving the accessibility of information in both existing and no-longer-existing urban places [171,172] or architecture [150]. The VALUE project [173] highlights user location, multimodal interaction, user understanding, and gamification as the four main pillars of innovative built use. The authors of [167,174] analyze the experiential aspect of in situ AR communication in cultural heritage and heritage management. The complexity of shapes and interaction with ambient light play a dominant role in AR visualization, increasing or decreasing the sense of presence [175]. Some studies investigate the impact of AR technology on the visitor's learning experience, deepening both cultural tourism and consumer psychology domains [176]. Even if many studies focused on MAR in the cultural tourism domain, few of them would explore the adoption of augmented reality smart glasses (ARSGs). Han et al. [177] contributed to highlighting the adoption of this technology in cultural tourism. The mobile capacity to merge locational instruments such as GPS and AR is helpful in guided tours and tourist storytelling, furthering the revivification of cultural heritage sites. Hincapié et al. [178] highlight the differences in learning and user perception with or without mobile apps, underlining the improvement in the cultural experience. This aspect has also been deepened by Han et al. [4], who studied the experiential value of AR on destination-related behavior. They investigated how the multidimensional components of AR's experiential value support behavior through AR satisfaction and experiential authenticity. In general, AR contributes to cultural dissemination and perception of tangible remains, while offering virtual reconstructions and complementary information, sounds, images, etc. That capacity, well known and applied in the cultural heritage field, can be enlarged in other domains and territories, i.e., attracting people to rural areas, and improving their social and economic situation [179];
- To visualize the reconstructive interventions or the simulations of intervention on the artifacts. This second domain is undoubtedly addressed to experts or technicians in the field, who are called to read the built environment and manage it in the best possible way, planning any intervention operations. This activity can be conducted at the urban scale by introducing decision-making and participatory processes—as the SquareAR project has shown [180]—or at the architectonic scale, showing the use of AR as a tool for reading the building to support the restoration process [181];
- To optimize and enrich the management and exploration of the monuments. This last strand is aimed at the managers or owners of the assets, using AR to manage

the heritage and its main features better, increasing its value in terms of visibility and induction [182]. Starting from the impact of the surrounding architectural heritage [183], some articles investigate the value of AR in providing a practical guide to the existing structures, increasing the accessibility and value of the asset [184]. Other research analyzes AR as a technology applied through devices and wearables for the promotion and marketing/advertising of tourist destinations [185], replacing the traditional tourist channels of explanation of monuments and architectures in situ [186].

### 8.2. AR for the Architectural Design and Building Process

The relationship between built architecture, design tools, and the use of AR technologies to simplify specific construction steps is a topic that began almost two decades ago and has found a very fertile research ground in architecture [187]. Today, augmented reality (AR) plays a relevant role in Industry 4.0, showing great potential within the construction industry [188]. One of the most important contributions in the architectural design domain is in representing a full-scale virtual model to visualize the project on-site [189]. This supports all actors involved in the process, from architects to interior designers, from structural designers to clients, employers, and workers. AR may support the entire design chain of a building, simplifying the reading, interpretation, and implementation of that complex intersection of activities expected in the AEC domain, and increasingly systematized by the lean and BIM processes. Some current states of the art [28,190] have highlighted the potential of AR in simplifying and optimizing that process and improving both the design and construction processes [191], emphasizing the integration between AR and BIM in the workflow [192]. Others have highlighted how AR can make BIM/CAD ubiquitous throughout the AR-BIM process, supporting and improving the management and budgeting of the artifact by analyzing critical consumption issues [193]. Augmented reality, therefore, is proving to be a supportive tool for every aspect of the construction life cycle:

- In project planning and co-design activities, the user can preview and share the building and urban context simulation. Urban/architecture planning requires a hierarchy of decision-making activities, enhanced, and optimized by collaborative interfaces. Co-design activities may involve both specialists and citizens, offering a decisive role in the urban and architecture pre-organization choices. Augmented reality can assume the role of a co-working platform. At the urban scale, AR is considered a practical design and collaboration tool within design teams. One of the first examples has been the ARUDesigner project, evaluating virtual urban projects in a real, natural, and familiar workspace [194]. Urban areas represent a meeting point between stakeholders' needs and public needs. For that reason, participatory processes are necessary, and AR can increase the range of participants and the intensity of participation, engaging citizens in collaborative urban planning processes [195]. At the architectural scale, before construction, virtual walks around the future buildings may identify changes that must be applied without causing problems and delays in the building execution phase. In this way, communication delays can be minimized, optimizing project management processes [196]. Some applications can overlay actual design alternatives onto drawings through MARs, allowing for flexible wall placement and application of interior and exterior wall materials, developing new, engaging architectural co-design processes [197]. Multitouch user interfaces (MTUIs) on mobile devices can be effective tools for improving co-working practices by providing new ways for designers to interact on the same project, from the cognitive and representation points of view [198];
- In the construction monitoring process: the real-time overlap between the design and the as-built model allows for sudden validation of the progress of construction. It can also highlight the inconsistencies between the design documentation, 3D model, and the actual state [199], assessing their effects and allowing for rapid response, reducing project risk. Structural elements can be monitored from the early construction stages

to their maintenance [190]. To monitor and control construction progress and perform comparisons between as-built and as-planned project data, quick access to easily understandable project information is necessary [200,201], presenting a model that overlays progress and planned data on the user's view of the physical environment. Moreover, in recent years RPAS platforms integrated with AR visualization are applied to the monitoring and inspection of buildings [202];

- In the reading and progress of AEC work, AR allows for supplemental or predictive information at the construction site—for example, by superimposing the structural layout with the architectural and plant systems in space, reducing errors during construction. An example is described by the D4AR project [203]. This ability to incorporate information on multiple levels can help workers to process complex information in order to document and monitor the in-progress construction [201]; it can also help workers to identify any on-site problems and avoid rework, quickly optimizing productivity;
- AR technology supports some phases of reading, construction, and assembly on-site—for example, by facilitating the reading of plans, construction details, and positioning of structural elements, minimizing errors caused by misinterpretation. [204]. Concerning distractions in on-site AR, emphasis should be placed on the careful use of this technology to avoid distractions caused by the screen and loss of awareness of the context in which one is immersed.

### 8.3. AR for Architectural Training and Education

This last section discusses AR as a tool for training different types of skills and users in architecture. What distinguishes the training aspect from the previous sections' informational aspects regards the learning activity of a specific task—repetitive or not—or contextual knowledge, scalable to different examples. The increasing use of AR for learning over the past two decades [205–207] comes to meet both the growing ability to use and manage digital data and the increasing need for dynamic content. The recent pandemic emergency has forced a sudden acceleration in the "remote learning" and "EdTech" domains. AR communication methodologies can propose new scenarios of augmented education in a constructive framework [36], verifying the learner's acquisition process (metacognition) through the interaction between the virtual and real environments. Moving to the transmitted AR content, Arduini [208] poses the need to modify it according to the new media. Consider, for example, the zSpace application, which imposes a reflection on the coherence between content and the way in which skills are transmitted. The implementation of MAR systems has undoubtedly facilitated AR in teaching, avoiding fixed technological infrastructure. Research on MAR in the education domain has increased significantly since 2013, showing the growing potential and performance of student learning. Dunleavy and Dede [209] both addressed AR as a learning tool and explored its educational and pedagogical roles in the classroom. Radu [210] compared AR and non-AR learning performance, listing advantages and disadvantages. In architecture, two main macro-areas of research are reported: education on the built heritage, and training for specific activities in the built domain. The former is required to understand the observed environment and artifact; the latter trains subjects in good construction, maintenance, and restoration practices on a given asset.

Heritage education involves all AR visualization of architectural models or, more generally, of cultural heritage, for educational purposes. It emphasizes the greater communicative and enhancement potential in the use of this form of visualization. This kind of in situ AR learning activity can be defined as discovery-based learning [211]. Engaging in this sense is the state of the art in Europe in the use of apps for heritage education, as shown by Luna et al. [212], highlighting that AR remains mostly applied for heritage education instead of modern/contemporary architecture. The state of the art proposed by Ibañez-Etxeberria [213] also compares different projects related to promoting cultural context, highlighting the connections and users' preferences with the cultural heritage

analyzed, while Gonzàles Vargas et al. [214] focused their attention on learning motivation. The AR for architecture promotion and understanding overlaps and differs for only one purpose: the dissemination and visibility of the artifacts, and their understanding. In this sense, the possibility of visualizing and efficiently managing a 3D architecture through a target exploring its main morphological features favors geometrical and functional learning [18]. This can be extended to architecture that no longer exists, or to projects that have never been realized, where their representation helps further understanding of the logic of construction [215].

On the other hand, education and training in the AEC field present a significant amount of literature related to AR learning in construction, engineering, architecture, and built management [216,217]. Design education represents a step that develops the knowledge and skills necessary to start a critical operational path on new or existing architectures. The role of tangible AR (tangible user interface (TUI)) can support a remote interactive design by providing a link between tactile action and visualization according to collaborative learning [218]. Morton [219] suggests implementing AR systems in design studios to improve students' design learning, quickly evaluating the multiple solutions developed in a design walkthrough. Design visualization also assumes a key role in the educational process. The introduction of the first luminous planning tables [220], and their subsequent development through AR with SAR, contributes to supporting collaborative data visualization and simulation of boundary conditions to verify a project's feasibility. In the context of project presentation, the students can enhance architecture representations by integrating 2D drawings and interactive shapes via SDAR [221]; that condition can be extended to the interior design realm. The SARDE (spatial augmented reality design environment) project has shown how to use an SAR system to support design decisions, reducing the gap between the presentation and the project content [222]. Finally, it is important to underline the impact of AR systems in the review of architectural designs based on the user's perspectives, highlighting the capacity of AR to involve many people in the same activity, reinforcing a co-design approach [223].

Finally, regarding the training activities for building construction, Shirazi and Behzadan Amir [224] presented a collaborative, context-aware MAR tool to improve the knowledge paths in the civil engineering curriculum. Despite the apparent technological advancement in which AR plays a key role, its deployment in best practices for workforce management and project interpretation [225,226] is still limited. On-site manual or printed drawings for project representation and communication are preponderant, leading to possible errors in interpretation. Moreover, on-site AR training can even involve just some parts of the entire design process, developing some optimized activities that reduce errors in the interpretation and commissioning of the project [227]. Introducing such technology can provide awareness of 3D spatial interpretation of 2D drawings and assist overall project planning activities [228]. Some training applications also include training activities to develop workers' skills in assembling products and understanding on-site safety regulations and equipment operation [229].

## 9. Conclusions

This article proposes an overview of augmented reality in the architecture domain, reporting the process of an AR project from the design step to the application, highlighting the pros and cons of each step. The goal is to suggest non-fragmented storytelling accessible to non-expert users in a specific research field. In retracing this process, each step has been outlined in the architecture domain, highlighting the main applications and outcomes. The architectonic domain is highly varied, proposing many transdisciplinary types of research not framed in a homogeneous structure. For this reason, cultural heritage and archaeological heritage at the architectural scale are often reported and discussed. The panorama that emerges from this state of the art is a highly active research field spanning more than 20 years, promising exciting future research opportunities. The temporal conjunction of some events makes this even more evident.

On the one hand, technological and algorithmic development brings AR closer to all users, investing in both MAR and SAR techniques; this latter is helpful in the architectural field, preserving a shared, co-design workflow. On the other hand, the increasingly pervasive adoption of geo-localized apps, supported by an extended network and massive digitization that involves both the built environment and processes, creates a fertile ground for development and experimentation. Finally, market interest in AR/MR/VR is increasingly evident, fostered by the vital intersection with data mining, deep learning, and machine learning. The next generation of MAR applications will add value to numerous application domains, including architecture, improving the quality of the user experience. The highly versatile demands of such applications should be supported by cloud-based, edge-based, localized, and hybrid architectures. The high bandwidth, ultra-low latency, and massive connectivity offered by future 5G systems herald a promising future for AR applications in the architecture domain, and will lead to an AR democratization process, projecting a new generation of hybrid representations and fostering knowledge of the built environment.

**Funding:** This research received no external funding.

**Conflicts of Interest:** The author declares no conflict of interest.

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
