# Peer review of "AR in the Architecture Domain: State of the Art"

_applsci, doi:10.3390/app11156800_

Round 1

Reviewer 1 Report

First of all: This is a really impressing piece of work and it was a real pleasure for me to read.

I would tell this in advance to set frame for the following remarks:

  • Again, I highly appreciate the approach taken to structure a heterogeneous field like AR for architecture. This is mostly very well done. Both, structuring and language of the article are well to receive and understand.
  • The article clearly states that most research is done with regards to cultural heritage. Nether the less I would expect some more facets of AR for architecture (also already briefly mentioned in the article but maybe worth to increase/deepen a bit): e.g., co-design / urban planning / architectural design (there has been several EU H2020 projects on that topic, e.g. https://cordis.europa.eu/project/id/688873) or – as briefly mentioned in the application section – cultural tourism (6 EU H2020 projects currently running on this topic – some with regards to AR applications).
  • From my understanding the article may be rather a SOTA than a (literature) review ?. Nether the less, it would be interesting especially for chapter 8 about applications of AR in architectural planning & design to provide some information about how articles were searched & selected, how classification of the approaches described in the articles was performed etc.

Please let me also raise some small remarks:

  • Concerning Fig. 1: I suppose that this article based on an ERC proposal and it is a really nice idea for that to highlight relations to ERC fields. Nether the less, for this article it would need some more justification were this assess-/assignment comes from. In case Fig. 1 would base rather on personal assessment than on data (as e.g. keywords) I would tend to drop it.
  • Concerning the levels of knowledge transfer: Please provide a reference (p. 6).
  • I understand that this article is intended rather than a classification than for discussing specific devices. Nether the less the capability of the specific device e.g. with regards to position and orientation tracking is a highly influencing parameter and may be worth to be mentioned / discussed (in 5.1.1 and/or 5.2) g. with regards to the approaches / data fusion for markerless tracking at large scale (cf. for MS Hololens).

Reviewer 2 Report

Congratulations for the job of describing the overview of Augmented Reality in the field of Architecture very adequately.
The text reports the process of an AR project from the design step to the implementation.
The pros and cons of each step are evidenced. 
It fulfills the objective of suggesting a non-fragmented and accessible narrative for non-expert users in a specific research field. 
In the paper through the whole process each step in the field of Architecture realm has been declined.

The text is brilliant and well organized. No plagiarism or similarity with other documents has been detected.

But I am not sure that the description of the evolution of Augmented Reality in the last 20 years is a research article but rather a historical account, but I leave it to the editors to decide if it is.
